# Transformer-Based Maneuvering Target Tracking

**DOI:** 10.3390/s22218482

**Published:** 2022-11-04

**Authors:** Guanghui Zhao, Zelin Wang, Yixiong Huang, Huirong Zhang, Xiaojing Ma

**Affiliations:** 1School of Artificial Intelligence, Xidian University, Xi’an 710071, China; 2School of Electronic Confrontation, National University of Defense, Hefei 230037, China

**Keywords:** attention mechanism, maneuvering target tracking, recurrent neural network, transformer-based network

## Abstract

When tracking maneuvering targets, recurrent neural networks (RNNs), especially long short-term memory (LSTM) networks, are widely applied to sequentially capture the motion states of targets from observations. However, LSTMs can only extract features of trajectories stepwise; thus, their modeling of maneuvering motion lacks globality. Meanwhile, trajectory datasets are often generated within a large, but fixed distance range. Therefore, the uncertainty of the initial position of targets increases the complexity of network training, and the fixed distance range reduces the generalization of the network to trajectories outside the dataset. In this study, we propose a transformer-based network (TBN) that consists of an encoder part (transformer layers) and a decoder part (one-dimensional convolutional layers), to track maneuvering targets. Assisted by the attention mechanism of the transformer network, the TBN can capture the long short-term dependencies of target states from a global perspective. Moreover, we propose a center–max normalization to reduce the complexity of TBN training and improve its generalization. The experimental results show that our proposed methods outperform the LSTM-based tracking network.

## 1. Introduction

With the rapid development of the electronic information industry, target tracking technology has been increasingly used in the military and civilian fields. The target tracking task aims to estimate the state of the target based on data measured by sensors. It can be classified into maneuvering and non-maneuvering target tracking, where “maneuvering” refers to the case in which the target suddenly changes its motion state. For the tracking of maneuvering targets, the interactive multi-model (IMM) algorithm, which uses multiple models to fit complex motion states, is considered [1]. Therefore, many tracking algorithms proposed subsequently were based on the IMM [2,3,4]. However, IMM-based algorithms are associated with the mismatch problem between the set of models and the target motion states. Furthermore, when the motion state of the target changes, a specific number of observations must be accumulated, resulting in the model estimation delay problem [5].

The development of deep neural networks, especially recurrent neural networks (RNNs) with memory ability, provides novel ideas to solve the problems of IMM-based algorithms [6,7,8,9]. The RNN [10] and long short-term memory (LSTM) networks [11] can estimate the state from the observation at each time step [6,12]. Nevertheless, the LSTM and RNN can only process the input sequence sequentially, resulting in long-distance memory fading problems [9]. Thus, the LSTM and RNN may reduce the correlation between trajectory points at different locations, which subjectively influences the modeling of maneuvering states. In addition, trajectory datasets are usually collected in a fixed-range coordinate system and preprocessed with min–max normalization [8,13,14]. However, the same maneuvering state in the dataset may correspond to trajectories with different initial positions, which increases the complexity of network learning. Moreover, the fixed distance range reduces the generalization of the network.

In this study, to accurately model and estimate the states of maneuvering targets, we propose a transformer-based network (TBN). Specifically, our proposed network applies the transformer network as an encoder to extract global features of the observation sequence. Simultaneously, 1D convolutional networks are applied as a decoder to estimate the state sequence from the features. Compared with the LSTM network, which processes observations sequentially, the TBN associates the observations at all positions and applies an attention mechanism to model their dependencies [15]. Thus, the features of the observations can be represented independently without regard to their position in the sequence [16,17,18]. Therefore, the TBN has better feature representation and global memory ability than LSTM [18]. Moreover, a learnable positional embedding is added to the input of the TBN to explore the temporal features of the observation sequence. Finally, a novel center–max normalization is applied by the TBN to improve generalization. Compared with the min–max normalization, our proposed center–max normalization transforms the trajectories from a fixed to a relative coordinate system with the initial observation point as the origin. The experimental results demonstrate that center–max normalization considerably increases the generalization of the TBN to trajectories with different distance ranges. Furthermore, center–max normalization also promotes the tracking performance of the TBN by reducing the complexity of trajectory learning.

## 2. Problem Formulation

Based on the previous research on maneuvering target tracking [4,7,8,19], we mainly considered point targets tracked by radar in the X-Y plane. Meanwhile, the problem of target birth and death was not considered in this study. Therefore, we assumed that $z_{k}$ is the observation vector and $x_{k}$ is the state vector at the *k*th time step. Specifically, $x_{k}=\left[c_{x,k},c_{y,k},v_{x,k},v_{y,k}\right]$ denotes the coordinates and corresponding velocities in the two-dimensional scene, and $z_{k}=\left[\theta_{k},d_{k}\right]$ denotes the azimuth and distance of the radar observation.

We intend to build a maneuvering target tracking model based on a deep neural network. The input to the model is the observation sequence $z_{1:K}=\left\{z_{1},z_{2},\cdots ,z_{K}\right\}$, and the output is the estimated state sequence ${\hat{x}}_{1:K}=\left\{{\hat{x}}_{1},{\hat{x}}_{2},\cdots ,{\hat{x}}_{K}\right\}$, where *K* is the total number of time steps. Given that target tracking is a regression problem, we used the root-mean-squared error (RMSE) between the normalized ground-truth sequence $x_{1K}^{*}=\left\{x_{1}^{*},x_{2}^{*},\cdots ,x_{K}^{*}\right\}$ and the estimated sequence ${\hat{x}}_{1:K}^{*}=\left\{{\hat{x}}_{1}^{*},{\hat{x}}_{2}^{*},\cdots ,{\hat{x}}_{K}^{*}\right\}$ as the loss function [9] to evaluate the model: (1)$$Loss=\sqrt{\frac{1}{K}\sum_{k=1}^{K}{\left({\hat{x}}_{k}^{*}-x_{k}^{*}\right)}^{2}}.$$

In practice, obtaining a sufficient number of trajectories is difficult. Thus, we simulated segmented trajectories based on the state-space model (SSM) [20].

The SSM defines the state transition equation and observation equation as: (2)$$\left\{\begin{array}{c} x_{k}=Fx_{k-1}+n_{k}\\ z_{k}=h\left(x_{k}\right)+u_{k} \end{array}\right.$$
where *F* is the transition matrix and $n_{k}$ is the transition noise. *h* is the nonlinear observation, and $u_{k}$ is the observed noise.

In this study, two motion states were considered: constant velocity (CV) and constant turn (CT), as mentioned in [8]. The transition matrix of CV and CT is defined as: (3)$$\begin{array}{c} F_{CV}=\left[\begin{array}{cccc} 1 & 0 & \tau & 0\\ 0 & 1 & 0 & \tau \\ 0 & 0 & 1 & 0\\ 0 & 0 & 0 & 1 \end{array}\right] \end{array}$$
(4)$$\begin{array}{c} F_{CT}=\left[\begin{array}{cccc} 1 & 0 & \frac{\sin\left(w\tau \right)}{w} & \frac{\cos\left(w\tau \right)-1}{w}\\ 0 & 1 & \frac{1-\cos\left(w\tau \right)}{w} & \frac{\sin\left(w\tau \right)}{w}\\ 0 & 0 & \cos\left(w\tau \right) & -\sin\left(w\tau \right)\\ 0 & 0 & \sin\left(w\tau \right) & \cos\left(w\tau \right) \end{array}\right] \end{array}$$
where *w* is the turn rate of the maneuvering target and τ is the sampling interval of the observations. According to [21], the transition noise $n_{k}=\left[n_{c,k},n_{c,k},n_{v,k},n_{v,k}\right]$ is calculated from: (5)$$\left[\begin{array}{c} n_{c,k}\\ n_{c,k}\\ n_{v,k}\\ n_{v,k} \end{array}\right]=\left[\begin{array}{cc} \frac{\tau^{2}}{2} & 0\\ \frac{\tau^{2}}{2} & 0\\ 0 & \tau \\ 0 & \tau \end{array}\right]\cdot \left[\begin{array}{c} \alpha_{k}\\ \alpha_{k} \end{array}\right]$$
where $\alpha_{k}∼N\left(0,\sigma_{a}^{2}\right)$ is the Gaussian noise caused by the maneuvering acceleration with zero mean and standard deviation $\sigma_{a}$.

For radar tracking, $Z_{k}$ is defined as: (6)$$\begin{array}{c} \left[\begin{array}{c} \theta_{k}\\ d_{k} \end{array}\right]=\underset{h\left(x_{k}\right)}{\left[\underset{︸}{\begin{array}{c} \mathrm{arc}\tan\frac{c_{y,k}}{c_{x,k}}\\ \sqrt{c_{x,k}^{2}+c_{y,k}^{2}} \end{array}}\right]}+\left[\begin{array}{c} u_{\theta ,k}\\ u_{d,k} \end{array}\right]\\ u_{\theta ,k}∼N\left(0,\sigma_{\theta }^{2}\right),u_{d,k}∼N\left(0,\sigma_{d}^{2}\right) \end{array}$$
where $\sigma_{\theta }$ is the standard deviation of the azimuth and $\sigma_{d}$ is the standard deviation of the distance.

## 3. Proposed Model

In this section, we discuss the components of the TBN in detail. In Section 3.1, we introduce a trajectory normalization method named center–max normalization to improve generalization. In Section 3.2, the structure of the TBN is presented. In Section 3.3, we summarize the overall process of applying the TBN for maneuvering target tracking.

### 3.1. Center–Max Normalization

A trajectory of a maneuvering target is exhibited in Figure 1. The left of Figure 1 shows an observation sequence, which contains the distance and azimuth. To eliminate the dimensional difference between the observations, $z_{1K}$ in the polar coordinates are converted to ${\tilde{z}}_{1K}$ in the X-Y plane coordinates: (7)$$\underset{{\tilde{z}}_{k}}{\left[\begin{array}{c} {\tilde{z}}_{x,k}\\ \underset{︸}{{\tilde{z}}_{y,k}} \end{array}\right]}=\left[\begin{array}{c} d_{k}\cos\left(\theta_{k}\right)\\ d_{k}\sin\left(\theta_{k}\right) \end{array}\right].$$

Figure 1c shows a trajectory in the X-Y plane coordinates. The distance range and initial position of the targets may vary extensively; thus, we propose a center–max normalization mechanism to improve the generalization of the model and reduce the training complexity, as shown in Figure 2. This can be formulated as follows: (8)$${\tilde{z}}_{k}^{*}=\frac{{\tilde{z}}_{k}-{\tilde{z}}_{1}}{D_{\max}},k=1,\cdots ,K$$
where ${\tilde{z}}_{k}^{*}$ is the normalized observation at the *k*th time step, ${\tilde{z}}_{1}$ is the initial value of ${\tilde{z}}_{1:K}$, and $D_{\max}$ denotes the maximum distance that the targets can move within *K* time steps. In Equation (Equation 8), the observation sequence is normalized to [−1,1] by dividing by $D_{\max}$. Subtracting ${\tilde{z}}_{1}$, the observation sequence ${\tilde{z}}_{1:K}$ is represented in a relative coordinate system with ${\tilde{z}}_{1}$ as the origin. Benefiting from center–max normalization, the TBN only needs to focus on learning different maneuvers of the target without considering the influence of the initial position. Therefore, the tracking of maneuvering targets by the TBN is not limited by the detection range. Correspondingly, the ground-truth state sequence $x_{1:K}$ is normalized as follows: (9)$$\begin{array}{c} x_{1:K}^{*}=\frac{x_{1:K}-c_{x}}{X_{\max}}\\ c_{x}=\left[{\tilde{z}}_{x,1},{\tilde{z}}_{y,1},0,0\right]\\ X_{\max}=\left[D_{\max},D_{\max},V_{\max},V_{\max}\right] \end{array}$$
where $x_{1:K}^{*}$ is the normalized state sequence, $c_{x}$ is the centering vector corresponding to $x_{1:K}^{*}$, $\left[{\tilde{z}}_{x,1},{\tilde{z}}_{y,1}\right]$ is the position component of ${\tilde{z}}_{1}$, and $V_{\max}$ is the maximum speed of the simulation targets.

### 3.2. Proposed Network

In sequence modeling tasks, the LSTM network sequentially extracts features. However, the transformer network uses the self-attention mechanism to process input data in parallel, which can capture both local and global dependencies. Therefore, we innovatively introduced it to the target tracking task to comprehensively capture the internal law of target maneuvering. Our proposed TBN consists of positional encoding, N-stacked transformer encoder layers, and one convolutional decoder layer. Each transformer encoder layer contains multi-head self-attention, a feedforward fully connected network, and two residual connections after each of the previous blocks. For intuitive understanding, the entire architecture of the TBN is shown in Figure 3.

#### 3.2.1. Positional Encoding

In natural language processing tasks, the transformer network adds positional encoding to the input tokens to represent their relative or absolute positions in the sequence [15]. However, in this study, the input to the TBN is numeric. Therefore, the learnable positional encoding mentioned [22] is added to the input of the network as follows: (10)$$s_{1:K}^{*}\left[i\right]=\left\{\begin{array}{c} w_{i}{\tilde{z}}_{1:K}^{*}+\varphi_{i},\;\mathrm{if}\;i=0\\ F\left(w_{i}{\tilde{z}}_{1:K}^{*}+\varphi_{i}\right),\;\mathrm{if}\;1\leq i\leq E \end{array}\right.$$
where F is the sine function, $w_{i}$ and $\varphi_{i}$ are learnable parameters that map ${\tilde{z}}_{k}^{*}$ to an *E*-dimensional representation space, and $s_{1:K}^{*}\left[i\right]$ is the encoding result of the *i*th subspace.

#### 3.2.2. Multi-Head Self-Attention

Self-attention is the core of the TBN. First, the input encoding sequence $s_{1:K}^{*}\in R^{E\times K}$ of self-attention is linearly mapped into the sequences “query” (Q), “keys” (K), and “values” (V) as follows: (11)$$\begin{array}{c} Q=W_{Q}\cdot s_{1:K}\\ K=W_{K}\cdot s_{1:K}\\ V=W_{V}\cdot s_{1:K} \end{array}$$
where $W_{Q},W_{K}$, and $W_{V}\in R^{E\times E}$ are learnable matrices.

Furthermore, *Q*, *K*, and *V* are split into *M* subsequences along dimension *E*, and *M* attention heads are obtained by the interaction of the elements at any two positions in each subsequence: (12)$$head_{m}=\mathrm{soft}\max\left(\frac{Q_{m}^{T}K_{m}}{\sqrt{d_{M}}}\right)V_{m},m=1,\cdots ,M$$
where $Q_{m},K_{m},V_{m}\in R^{E_{m}\times K}$ and $E_{m}=\frac{E}{M}$.

Finally, *M* attention heads are concatenated to compose the multi-head self-attention: (13)$$s_{attention}=Concat\left(head_{1},\cdots ,head_{M}\right).$$

Thus, the network is allowed to capture more information from different representation subspaces at different positions.

#### 3.2.3. Feedforward Layer

After the multi-head self-attention, a feedforward layer consisting of two fully connected layers is used to linearly transform each position of $s_{attention}$.

In the decoder part, two 1D convolutional layers are used to output the final trajectory estimation ${\hat{x}}_{1:K}$, and the parameters of the network are trained by minimizing Equation (Equation 1) using the mini-batch gradient descent.

### 3.3. Maneuvering Target Tracking Based on the TBN

When the well-trained TBN is applied to track a complete trajectory, all observations are first segmented with window length K=10 and step size P=5. These segmented observation sequences are then normalized sequentially and passed to the TBN to estimate the corresponding state sequence set $\left\{{\left({\hat{x}}_{1:K}^{1+rP}\right)}^{*},r\in \left(0,\cdots ,R-1\right)\right\}$, where ${\left({\hat{x}}_{1:K}^{1+rP}\right)}^{*}$ denotes the normalized state sequence output at time step (1+rP) and *R* is the number of sequences. Subsequently, ${\left({\hat{x}}_{1:K}^{1+rP}\right)}^{*}$ needs to be denormalized as follows: (14)$${\hat{x}}_{1:K}^{1+rP}={\left({\hat{x}}_{1:K}^{1+rP}\right)}^{*}⊙X_{\max}+c_{x}^{r},r=0,\cdots ,R-1$$
where $c_{x}^{r}$ is the centering vector of the *r*th state sequence. In addition, adjacent state sequences ${\hat{x}}_{1:K}^{1+rP}$ and ${\hat{x}}_{1:K}^{1+\left(r+1\right)P}$ are merged together. Let ${\bar{x}}_{1:2K-P}^{1+rP}$ denote the merge result of two above-mentioned state sequences, whose length is 2K−P. Thus, the overlapped regions of ${\bar{x}}_{1:2K-P}^{1+rP}$ are calculated as follows: (15)$${\bar{x}}_{P:K}^{1+rP}=0.5\left({\hat{x}}_{P:K}^{1+rP}+{\hat{x}}_{1:K-P}^{1+(r+1)P}\right).$$

Finally, all state sequences in the set $\left\{\left({\hat{x}}_{1:K}^{1+rP}\right),r\in \left(0,\cdots ,R-1\right)\right\}$ are merged in turn to obtain complete state estimates. Figure 4 illustrates the overall tracking process, including the observations segmentation, center–max normalization processing, network estimation, denormalization processing, and segmented state sequences concatenation.

## 4. Experiments and Results

In this section, we list the parameters of the trajectory dataset and the TBN. Several experiments were designed to test the tracking performance of our proposed model.

### 4.1. Implementation Details

**Dataset**: We generated 300,000 trajectories based on the SSM as a dataset. The parameters of the trajectory dataset are listed in Table 1. In addition, we assumed normalization parameters: $D_{\max}=3$ km, $V_{\max}=300$ m/s, and targets were observed every 1 s.

**Hyper–parameters**: Our network consists of four encoder layers, with eight attention heads. The dimension of *E* was 512. The output dimensions of the 1D convolutional layer in the decoder were 64 and 4, respectively. The model was trained using the Adam optimizer [23] with $\beta_{1}=0.9,\beta_{2}=0.98$, and $\varepsilon =10^{-9}$. The learning rate was linear warmed-up for the first 10 epochs and decayed subsequently based on the dynamic adjustment strategy mentioned in [15]. We trained 300 epochs with a batch size of 64 on a single NVIDIA TITAN Xp GPU.

**Baseline**: We compared the TBN+center–max normalization (TBN+CM) model with the IMM algorithm [19] and the LSTM+min–max normalization (LSTM+MM) tracking model [8]. As a comparison, we also built the LSTM+center–max normalization (LSTM+CM) model. The LSTM network consisted of four hidden layers with a dimension of 128, as mentioned in [8]. The same dataset was used to train the above networks.

### 4.2. Results

We first compared the performances of the LSTM+MM, LSTM+CM, and TBN+CM based on a test set containing 20,000 segmented trajectories. The tracking results are listed in Table 2. In Table 2, the position and velocity RMSEs of the LSTM+CM are smaller than that of the LSTM+MM, which proves that our proposed center–max normalization improved the tracking capability of the network by reducing the complexity of trajectory learning. At the same time, the TBN+CM achieved the smallest position and velocity RMSE. Thus, it can be concluded that the TBN yields better performance than LSTM when tracking segmented trajectories.

We then simulated a target with the initial states of [2 km, 2 km, 50 m/s, 0 m/s] and steering rates equal to $0^{\circ }$ and conducted Monte Carlo simulations to generate a 60-step trajectory named A1. The target maneuvers had turn rates equal to $-1^{\circ }$ and $3^{\circ }$ at the 10th step and the 40th steps, respectively. In addition, the standard deviations of acceleration, azimuth, and distance noise were set to 5 m/s${}^{2}$, 0.2${}^{\circ }$, and 5 m. We evaluated the TBN+CM, LSTM+MM, LSTM+CM, and IMM algorithms on trajectory A1. The tracking results are listed in Table 3 and Figure 5.

Among the listed figures, Figure 5a shows how well the algorithms tracked the target. Figure 5b,c show the pointwise RMSE of trajectory A1. Furthermore, the average RMSEs of trajectory A1 are listed in Table 3. In Table 3, the RMSEs of the LSTM+CM are smaller than those of the LSTM+MM, which proves that our proposed center–max normalization improved the tracking capability of the network by reducing the complexity of trajectory learning. At the same time, the bolded results in Table 3 indicate that TBN+CM had the smallest tracking error. The experiments above demonstrated the superiority of the TBN+CM in tracking maneuvering targets.

In addition, the initial position of trajectory A1 was moved to [12 km, 12 km] and [15 km, 15 km] to obtain trajectories A2 and A3. We conducted generalization experiments on trajectories A2 and A3, as listed in Table 4. The bolded results in Table 4 demonstrated that our proposed TBN+CM can generalize to tracking trajectories beyond the preset distance. However, the LSTM+MM led to tracking failure due to its fixed normalization mechanism.

## 5. Conclusions

In this study, we employed the attention mechanism of the transformer network to extract a comprehensive tracking of trajectories and finally developed a novel network named the TBN for radar target tracking missions. Furthermore, our proposed center–max normalization improved the generalization of the network by processing observations in a relative coordinate system. It can be seen from the experimental results that, when tracking maneuvering targets, our proposed TBN model obtained lower RMSEs of position and velocity than the LSTM model, and the TBN model can still work normally when the observation sequence is missing; however, the LSTM model will not be available. Therefore, our algorithm outperformed existing LSTM-based tracking networks and traditional algorithms.

## Figures and Tables

**Figure 1 sensors-22-08482-f001:**
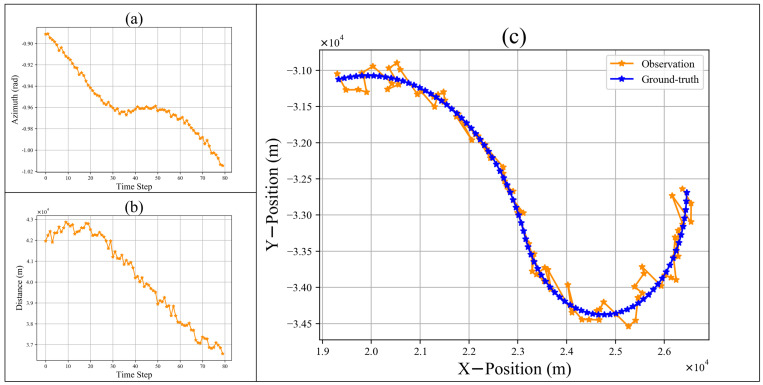
The observation and ground-truth of a trajectory. The (**a**) is the azimuth observation sequence of the trajectory. The (**b**) is the distance observation sequence of the trajectory. The (**c**) is the observation and ground-truth sequence in the X-Y plane coordinate system.

**Figure 2 sensors-22-08482-f002:**
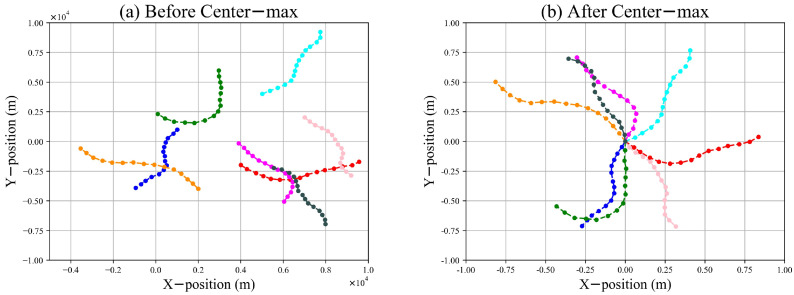
Center–max normalization. After center–max normalization, the distance ranges of the trajectories are transformed to [−1,1] and the differences in the initial positions of the trajectories are removed.

**Figure 3 sensors-22-08482-f003:**
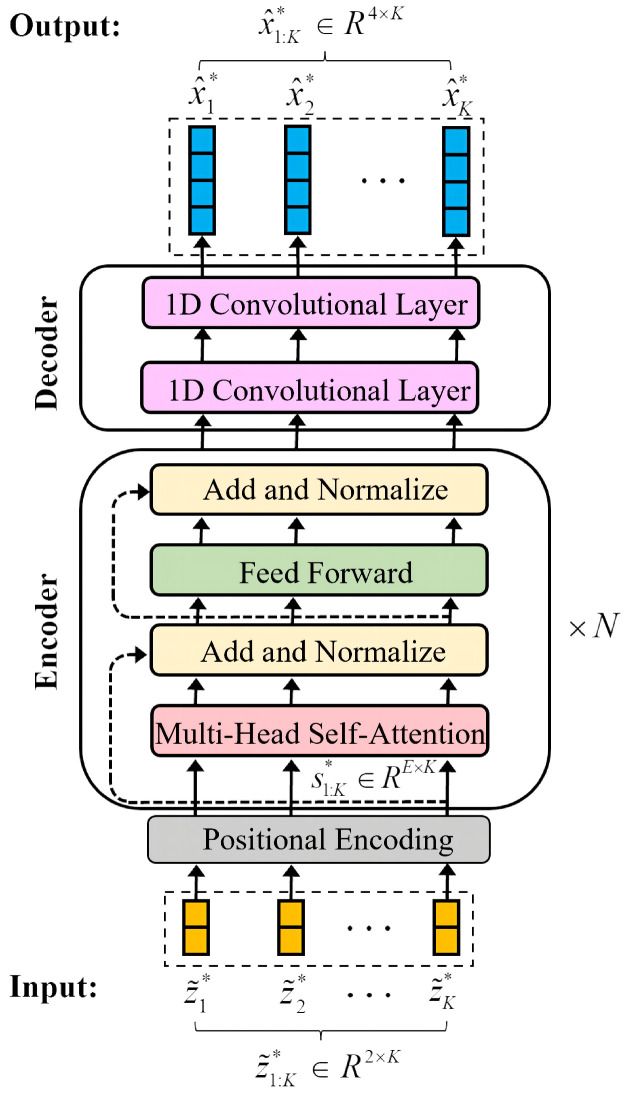
Architecture of the TBN. Input data are normalized observation sequences ${\tilde{z}}_{1:K}^{*}$. ${\tilde{z}}_{1:K}^{*}$ is first mapped to $s_{1:K}^{*}$, whose dimension is E×K by positional encoding. The encoder consists of N-stacked multi-head self-attention and fully connected feedforward layers, which aim at extracting the features of $s_{1:K}^{*}$. The decoder maps *E*-dimensional feature vectors to the normalized state sequence ${\hat{x}}_{1:K}^{*}$ by two 1D convolutional layers.

**Figure 4 sensors-22-08482-f004:**
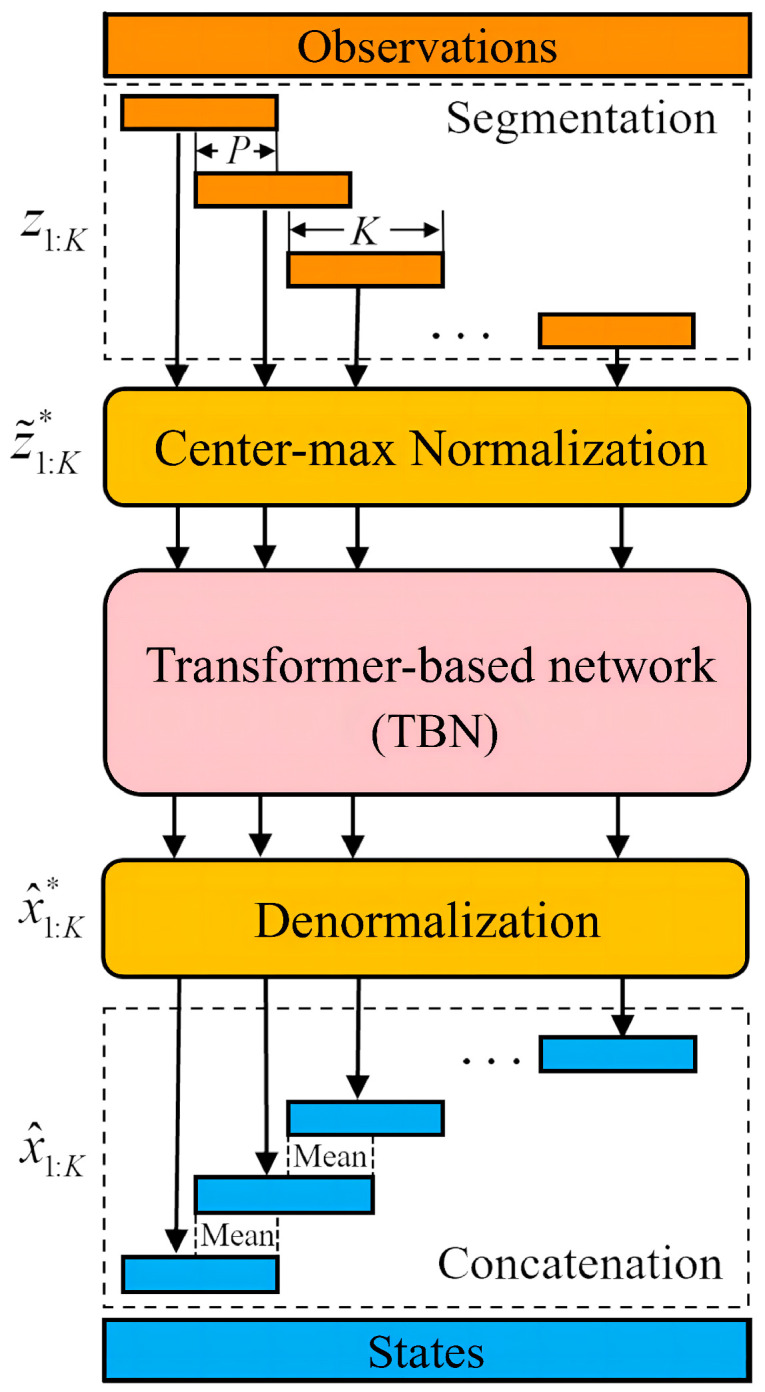
Structure of the transformer-based maneuvering target tracking. The observation sequences of targets are firstly segmented into subsequences $z_{1:K}$ of length *K* with step size *P*. After that, $z_{1:K}$ are converted to ${\tilde{z}}_{1:K}^{*}$ by center–max normalization. Then, the TBN infers the normalized trajectory ${\hat{x}}_{1:K}^{*}$ from ${\tilde{z}}_{1:K}^{*}$. In addition, ${\tilde{z}}_{1:K}^{*}$ are de-normalized to ${\hat{x}}_{1:K}$. Finally, the overlapped region of ${\hat{x}}_{1:K}$ is averaged and concatenated to obtain the estimation of the complete state sequences.

**Figure 5 sensors-22-08482-f005:**
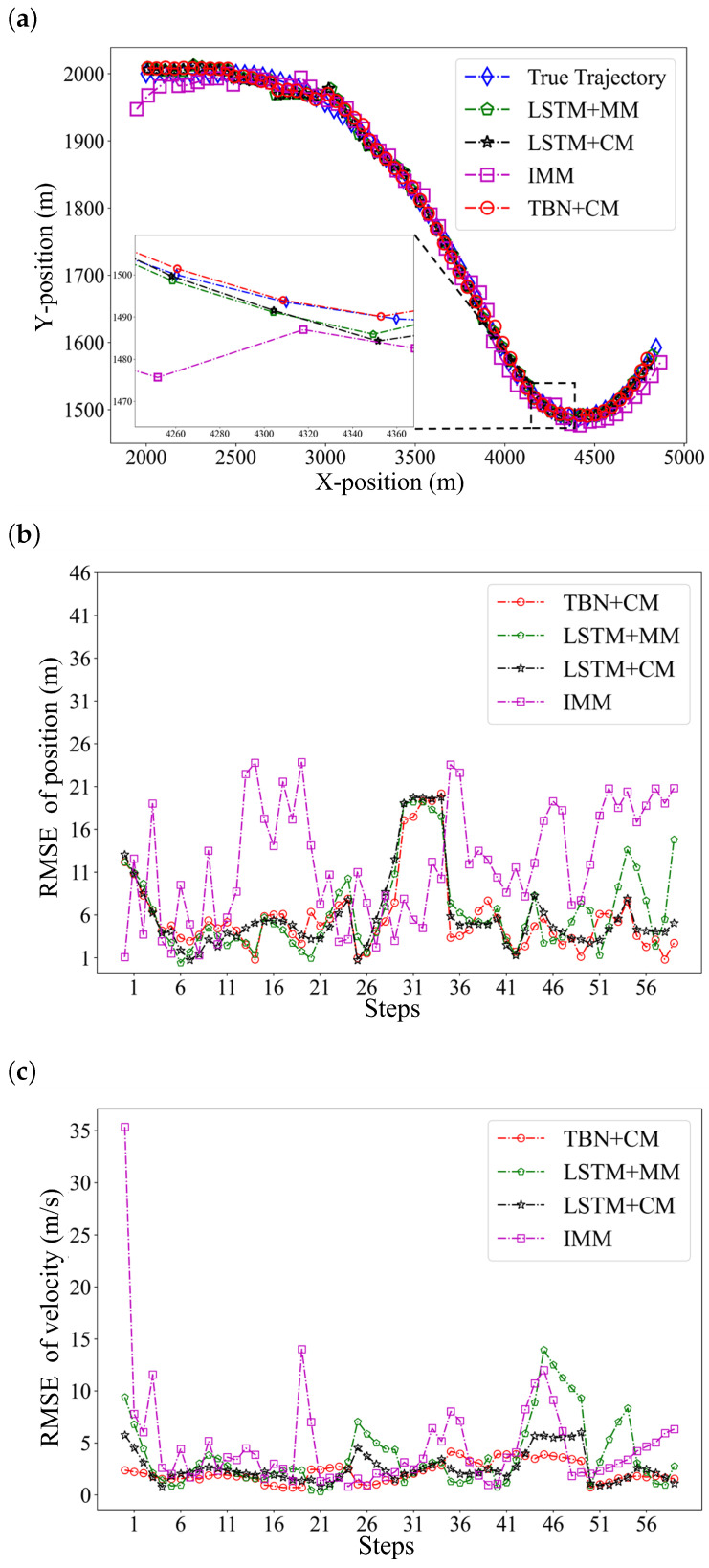
The result of tracking a maneuvering target by the TBM+CM, LSTM+MM, LSTM+CM, and IMM algorithms. (**a**) Tracking trajectory in the X-Y plane. (**b**) Pointwise position RMSE. (**c**) Pointwise velocity RMSE.

**Table 1 sensors-22-08482-t001:** Parameters of the trajectory dataset.

Parameter	Value
Distance range	1km∼10km
Angle range	$0^{\circ }$∼$360^{\circ }$
Velocity range	−300m/s∼300m/s
Turn rate (w)	$-10{\;}^{\circ }/s$∼$10{\;}^{\circ }/s$
The standard deviation of acceleration noise $\left(\sigma_{a}\right)$	$2\;{m/s}^{2}$∼$8\;{m/s}^{2}$
The standard deviation of azimuth noise $\left(\sigma_{\theta }\right)$	$0.1^{\circ }$∼$0.3^{\circ }$
The standard deviation of distance noise $\left(\sigma_{d}\right)$	5m∼8m

**Table 2 sensors-22-08482-t002:** Numerical results of several methods for tracking segmented trajectories.

	RMSE of Position (m)	RMSE of Velocity (m/s)
LSTM+MM	16.27	6.75
LSTM+CM	14.43	5.14
**TBN+CM**	**13.50**	**3.64**

**Table 3 sensors-22-08482-t003:** Numerical results of several methods for tracking trajectory A1.

	RMSE of Position (m)	RMSE of Velocity (m/s)
IMM	14.54	6.35
LSTM+MM	11.82	4.64
LSTM+CM	10.30	3.47
**TBN+CM**	**9.33**	**2.04**

**Table 4 sensors-22-08482-t004:** Results of tracking trajectories at different initial positions.

	RMSE of Position (m)		RMSE of Velocity (m/s)	
	**TBN+CM**	**LSTM+MM**	**TBN+CM**	**LSTM+MM**
A2	**9.94**	146.14	**2.08**	56.19
A3	**9.15**	295.71	**2.03**	78.92

## Data Availability

Not applicable.

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
