# Peer review of "Transformer-Based Maneuvering Target Tracking"

_sensors, 2022, doi:10.3390/s22218482_

Round 1
Reviewer 1 Report
1. Candidates have propose a transformer-based network to accurately model and estimate the states of maneuvering targets.
2. proposed network applies the 48 transformer network as an encoder to extract global features of the observation sequence.
3. 1-D convolutional networks are applied as a decoder to estimate the state 50 sequence from the features.
4. This work employed the attention mechanism of the transformer network to extract a comprehensive cognition of trajectories, and finally develop a novel network named TBN for radar target tracking missions. Furthermore, our proposed center-max normalization improved the generalization of the network by processing observations in a relative coordinate system.
5. I feel to consider this manuscript for publication in this esteemed journal as in its current form.
Author Response
Response:
We thank the reviewer for reading our paper carefully and giving the above positive comments.
Reviewer 2 Report
see attached file

Author Response
Response:
see attached file

Reviewer 3 Report
Below are my notes.
1) Please insert a space before each reference.
2) L. 79: Please check the h(·) symbol.
3) Please insert a space between the quantity and its respective unit in Figs. 1,2 and 5.
4) Please insert a period after Eqs. 1, 7, 13 and 15.
5) L. 99: After "Figure 2", replace the comma with a period.
6) Eq. 10: Please insert a space after "if" (bottom and top).
7) L. 168: Please insert a space between the quantity and its respective unit. Also, I recommend writing km and m/s (not m / s) as text and not as an equation. The same for Table 1. By the way, is the ˘ symbol really the most appropriate?
8) Table 2: *m/s.
9) I suggest joining the first two paragraphs of section 4.2.
10) Lines 191, 195, 206: It would be more appropriate to write the units as text and insert a space after each quantity. *m/s and *m/s^2.
11) Tables 3 and 4: *Position (m) and Velocity (m/s).
12) In the last section, please insert an explanation as to why the authors concluded that "The experimental results showed that our algorithm outperformed existing LSTM-based tracking networks and traditional algorithms.". That is, why did the proposed model present better results than the others?
Author Response
Response:
see attached file.

Round 2
Reviewer 2 Report
No further comments